# The KCL-SAIR team's entry to the GENEA Challenge 2023 Exploring Role-based Gesture Generation in Dyadic Interactions: Listener vs. Speaker

Viktor Schmuck, Nguyen Tan Viet Tuyen, Oya Celiktutan
Centre for Robotics Research, Department of Engineering, King's College London
London, UK
{viktor.schmuck;tan_viet_tuyen.nguyen;oya.celiktutan}@kcl.ac.uk

## ABSTRACT

This paper presents the KCL-SAIR team's contribution to the GENEA Challenge 2023. As this year's challenge addressed gesture generation in a dyadic context instead of a monadic one, our aim was to investigate how the previous state-of-the-art approach can be improved to be more applicable for the generation of both speaker and listener behaviours. The presented solution investigates how taking into account the conversational role of the target agent during training and inference time can influence the overall social appropriateness of the resulting gesture generation system. Our system is evaluated qualitatively based on three factors, including human likeness, appropriateness for agent speech, and appropriateness for interlocutor speech. Our results show that having separate models for listener and speaker behaviours could have potential, especially to generate better listener behaviour. However, the underlying model structures between the speaker and listener behaviour should be different, building on previous state-of-the-art monadic and dyadic solutions.

## CCS CONCEPTS

• **Human-centered computing** → **HCI theory, concepts and models**; *Empirical studies in interaction design*; **User studies**.

## KEYWORDS

datasets, Tacotron2, gesture generation, dyadic interaction

**ACM Reference Format:**
Viktor Schmuck, Nguyen Tan Viet Tuyen, Oya Celiktutan. 2023. The KCL-SAIR team's entry to the GENEA Challenge 2023 Exploring Role-based Gesture Generation in Dyadic Interactions: Listener vs. Speaker. In *Proceedings of 25th ACM International Conference on Multimodal Interaction (ICMI'25)*. ACM, New York, NY, USA, 6 pages. https://doi.org/XXXXXXX.XXXXXXX

## 1 INTRODUCTION

The generation of non-verbal behaviours in order to accompany the speech of both embodied agents and social robots enhances their perceived acceptance. Due to its importance, there has been a

*ICMI'25, October 2023, Paris, France*
© 2023 Association for Computing Machinery.
ACM ISBN 978-x-xxxx-xxxx-x/YY/MM...$15.00
https://doi.org/XXXXXXX.XXXXXXX

growing effort related to this line of research in the past years [1, 9, 10, 13, 17, 23, 25]. Agents employing gestures during communication allows them to add emphasis to the information they convey and to express their intentions or emotions. It is important to differentiate between monadic and dyadic settings when generating behaviours. In a monadic setting the agent exists alone, while in a dyadic one, its behaviour should be related to an interlocutor's, as it participates in a dynamic exchange taking turns speaking and listening.

Previous work established a state-of-the-art approach for generating gestures in a monadic setting based on an agent's speech and text information [9]. To extend this method to a dyadic setting, the interlocutor's verbal and non-verbal signals should also be taken into account. However, the listener and speaker behaviours of agents are significantly different [2, 6]; the listener is much more passive and occasionally mimics the speaker gestures with delayed synchrony. Therefore, this problem could benefit from a split training approach, where gesture generation in a dyadic context is broken down into listener and speaker behaviours.

Motivated by the importance of gesture generation for both virtual and embodied agents and the stark difference between listener and speaker behaviours in a dyadic context, this paper investigates the effect of training and employing multiple gesture generation models based on the speaker status of the agent. The qualitative assessments of our contribution show that compared to the simple dyadic extension of previous state-of-the-art [9], this technique is on par with several model improvement based techniques and the previous baseline.

## 2 BACKGROUND AND PRIOR WORK

In recent years there has been a growing interest in the research area of co-speech gesture generation for virtual [3, 5, 27] and embodied [22, 24] agents. The approach to gesture generation can be divided into two groups: rule-based [8] and data-driven approach [13, 24]. With the rule-based approach, the association of text or speech and gestures are pre-defined by a set of rules [8]. Consequently, this approach can only produce gestures in pre-designed contexts. With the data-driven approach, the relationship between gestures and text or speech is captured by end-to-end learning frameworks. Several studies used an Encoder-Decoder [5] architectures, Generative Adversarial Networks (GANs) [22, 26, 27] or Conditional GANs (cGAN) which were designed with Convolutional Neural Networks [24].

To foster the development of more appropriate gesture generation, GENEA Challenge 2023 [14] provides a dataset and a platform to create and evaluate non-verbal behaviour generation solutions.

The organisers provided a refined and split dataset based on the Talking With Hands 16.2 M [16] data. Moreover, they provided a baseline model [9, 18] which was adapted from the monadic gesture generation winner of a previous year's challenge.

In dyadic interaction, an essential aspect of co-speech gestures is the dynamic exchange of non-verbal signals between two partners for adapting to interacting social norms [15] and building a common ground [19]. As a result, the work presented in this paper will shed light on this important aspect. Specifically, our solution described below builds upon the baseline provided for the challenge [9] and investigates the effect of training separate speaker and listener gesture models. This approach is supported by the work of Alibali et al. [2] and Binder [6] who explored the non-verbal behaviour of speakers and listeners in a conversation. Alibali et al. [2] state that the listener behaviour can be limited to back-channel feedbacks such as nodding, saying "uh-huh", and occasional head movement indicating that something is not clear. Similarly, Binder [6] found that listeners also exhibit behavioural synchrony which plays a significant role in the positive perception of conversation partners. Based on their research, due to the stark difference between the behaviour expected from speakers and listeners, we believe the training of separate speaker and listener models is a promising avenue.

## 3 DATA AND DATA PROCESSING

The solution presented in this paper is using the training and validation sets of the Talking With Hands 16.2 M dataset presented by Lee et al.[16]. Using the same training and validation practices of the monadic motion generation solution proposed by Chang et al.[9], our solution utilises the speaker identity, text, audio, and motion information of the main-agent. In addition, the interlocutor's text, audio, and motion information is also used in order to extend the baseline to a dyadic setting.

Following the preprocessing practices presented by Chang et al. [9], we produce a mel spectrogram and MFCC features, as well as audio prosody features such as audio intensity, pitch, and their derivatives. To process text data, a FastText word embedding [7] is generated with 300 dimensions. As for the motion input data, we use the joint angle information provided in the dataset and extract information for 25 joints, 19 and 6 for the upper- and lower-body joints respectively. The joint angles were parameterised with exponential map [11]. Finger motion data was not used due to its reliability in the dataset, and we also use a root position of the body, resulting in $26 * 3 = 78$ features, 3 dimensions (i.e., 3D orientation information) for each joint information. This feature engineering was kept consistent with the one described by the state-of-the-art in order to provide a reliable comparison to the baseline method of Chang et al.[9] and to observe the direct effect of our sample selection method described below.

## 4 METHOD

Our method is primarily based on the baseline method proposed by Chang et al. [9]. This solution used a Tacotron2 [21] based architecture that was aimed to align speech features with gestures. This sequence-to-sequence approach was extended to use the interlocutor's motion, audio, text, and *speaker identity* features as inputs to

appropriate it to a dyadic context. Due to the increased input size, the original model's [9] hyperparameters were individually tuned as described in the challenge description paper [14].

Regarding our core contribution, we introduced the training of two separate models, constructed based on the baseline model structure. When training and validating the models, in one case, when selecting training samples, the *speaker identity* labelling of the agent was used to determine when the agent was speaking. This information was acquired from the dataset by concatenating the text input with the *speaker_id* using the same sample generation pipeline as the IVI baseline did [9]. If the sampling window yielded a non-zero sum of the resulting feature array, the agent was considered speaking. If both the main agent and the interlocutor were speaking, we consider the agent as 'speaking'.

Only training samples with speech were used to train a speaking model (SM) which was validated on samples where the agent was speaking. Similarly, our second model was trained and validated solely on samples where the *speaker identity* indicated that the agent was listening, resulting in a trained listener model (LM). In the dataset, there are some instances when both the agent and the interlocutor are speaking. These samples were used to train the SM, as the required gestures should still be appropriate for the agent providing expressions to support its speech.

The models were trained on the training set of the Talking With Hands 16.2M dataset [16] with the same hyperparameters as established in the challenge description paper [14], however, the batch size was reduced to 32 from the original 64 due to computational constraints. The full parameter list can be found at [18] in *Tacotron2/common/hparams_dyadic.py*.

Our models were trained until convergence, as stated in [18], around 20 to 30 thousand epochs. SM converged after $28k$ iterations, while LM converged by the $30k$ mark.

The training was performed on a Dell XPS 15, i9-13900H (14 cores, up to 5.40GHz Turbo), 32GB RAM, NVIDIA GeForce RTX 4070 - 8GB. Training and validation took around 16 and 18 hours for the SM and LM respectively.

During inference, both models were loaded into the 'generate_all_gestures.py' script provided by [18]. Outputs were generated frame by frame, selecting SM or LM depending on the *speaker identity* of the agent as described above. The resulting outputs were converted to joint angles utilising the built-in functions provided by the evaluation script.

A representation of the training and testing of the proposed models can be seen in Figure 1. The source code of our solution, adapted from the [18] repository can be found at [20].

## 5 EVALUATION

The evaluation was performed with the other GENEA Challenge 2023 submissions by the organisers as presented in [14]. The provided test set was formatted the same way as the training and validation sets of the Talking With Hands 16.2 M dataset [16] with the exception of the agent not having the motion samples for this split. The agent gestures were generated as described above, in Section 4.

Due to the lack of ground truth data, features such as Average Precision Error (APE), difference in Acceleration, and Jerk were not

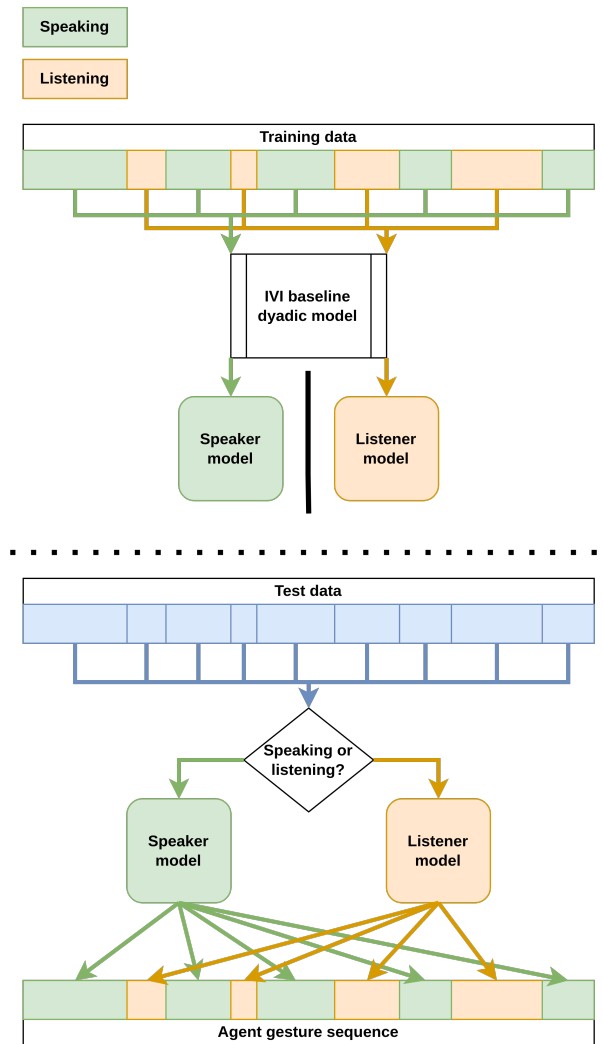

**Figure 1: A representation of how *speaker identity*-based sampling was introduced to train and test two models: one trained on speaker data; and one trained on listener data.**

measured. Instead, the resulting dyadic gestures were evaluated with regard to Human Likeness, Appropriateness for agent speech, and Appropriateness for the interlocutor in a large-scale crowd-sourced subjective evaluation. Human likeness measures whether the generated gesture resembles real human gestures. The appropriateness of the agent and interlocutor speech evaluations measure whether the generated gestures look natural with regard to the respective speaker. In the following sections, they are also referred to as *monadic* and *dyadic appropriateness*. Notably, appropriateness scores were measured by pairing the gestures generated for the correct speech segments, but also by pairing and showing mismatched speech-gesture stimuli pairs to participants.

For further details regarding the evaluation please refer to the main Challenge description paper [14].

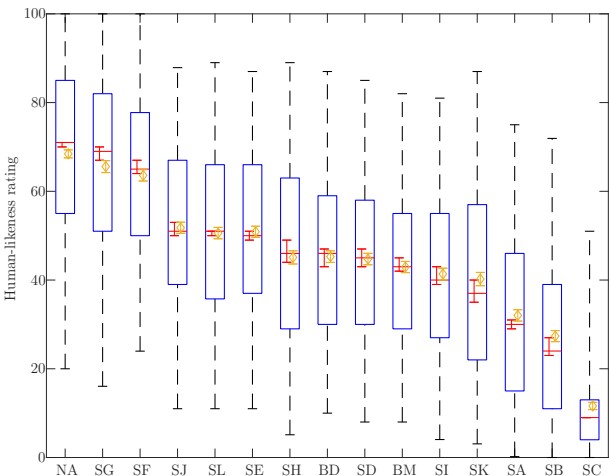

**Figure 2: Box plot visualising the ratings distribution in the human-likeness study. Red bars are the median ratings (each with a $0.05$ confidence interval); yellow diamonds are mean ratings (also with a $0.05$ confidence interval). Box edges are at $25$ and $75$ percentiles, while whiskers cover $95\%$ of all ratings for each condition. Conditions are ordered by descending sample median rating.**

## 6 RESULTS AND DISCUSSION

This section reports the three aspects of the qualitative evaluation performed on our solution. In the following sections, the proposed solutions will be labelled **SA-SL**, the baseline method's [9] monadic version is labelled **BM**, and the dyadic **BD**. Finally, the ground truth gestures recorded in the original dataset [16] are labelled **NA** (i.e., natural). Our proposed solution is labelled **SD**.

### 6.1 Human Likeness

Based on the responses of 200 participants, the median ratings between different conditions were analysed based on Mann-Whitney U tests, which is an unpaired non-parametric test. After acquiring the p-values, they were adjusted for multiple comparisons with the Holm-Bonferroni method [12].

The rating distribution of the human-likeness test and the significance of pairwise differences between conditions can be seen in Figure 2 and 3 respectively.

Based on the results, only 12 condition pairs out of the overall 105 were significantly different at $\alpha = 0.05$. Regarding our solution, its conditions were not different from other generated gestures in the set of {BD, BM, SD, SH}. However, they were statistically different from the set of {SE, SJ, SL}. This means that our solution achieved the same human likeness scores as SH, and the dyadic and monadic baselines. Finally, it was rated better with regard to human-likeness compared to SA, SB, SC, SI, and SK.

Based on these results, specifically examining human-likeness, we can say that our proposed approach does not hinder performance compared to the benchmarks. However, using speaker and listener models alone is not enough in a dyadic setting, as indicated by the significantly better-performing set of {SE, SF, SG, SJ, SL} models, and

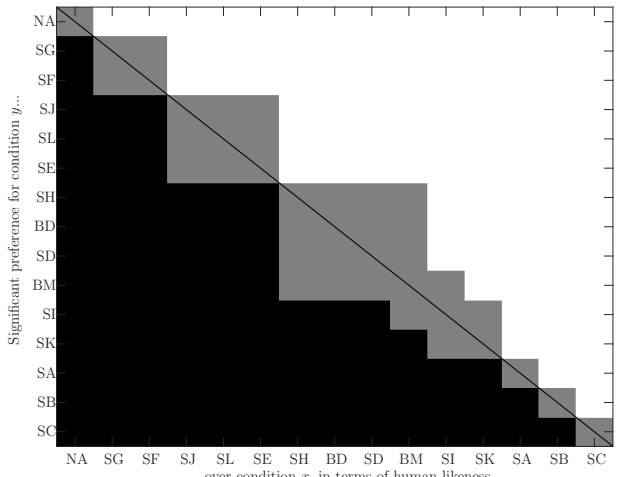

**Figure 3: Significance of pairwise differences between conditions. White means that the condition listed on the $y$-axis rated significantly above the condition on the $x$-axis, black means the opposite ($y$ rated below $x$), and grey means no statistically significant difference at the level $\alpha = 0.05$ after Holm-Bonferroni correction [12]. Conditions are listed in the same order as in Figure 2.**

the significantly higher mean and median human-likeness scores of the ground truth.

## 6.2 Appropriateness for agent speech

The appropriateness for agent speech (i.e., monadic appropriateness) was evaluated with 600 participants who contributed 36 ratings to this part of the study, with every condition receiving at least 1766 scores. The scores represent a mean appropriateness score (MAS), which is calculated by converting user responses to a 5-point scale ranging from $-2$ to $2$. The MAS are shown in Table 1(a) and represented in Figure 4(a). Furthermore, similar to the human-likeness evaluation, the pairwise comparison of solutions can be seen in Figure 5(a). To compare the performance of different solutions, Welch's $t$-test, an unpaired statistical test was used. To correct the test results for multiple comparisons a technique called the BH non-adaptive one-stage linear step-up procedure [4] was used.

Based on the results, our solution is statistically different from chance level performance (see dashed line in Figure 4(a)). The natural (NA) condition was significantly more appropriate compared to all synthetic condiditons. Regarding the condition of our proposed solution (SD), it was significantly more appropriate than the condition sets of {SC, SL} and {SA, SB, SH}. Moreover, it was not significantly different from the condition set of {BD, SE, SI, SK}. The remaining 7 conditions and NA were found to be significantly more appropriate than SD. As for preference comparison, SD was significantly more preferred than SC and SL, and it was less preferred than conditions NA, SG, and SJ.

Furthermore, we can infer that our proposed solution can match other conditions with regard to user preference when it comes to monadic appropriateness. However, it fails to be distinguished from

BD. This might be due to BD being trained on all available samples, while SD is only trained for dyadic cases on samples where the agent is speaking. It could be that with an equal number of training samples, its performance would show significant improvement. However, it seems approaches focusing on model improvements can improve monadic appropriateness more reliably.

## 6.3 Appropriateness for interlocutor speech

The appropriateness for interlocutor speech (i.e., dyadic appropriateness) was evaluated with 600 participants who contributed 36 ratings to this part of the study, with every condition receiving at least 993 scores. Just as in the case of the monadic appropriateness evaluation, the scores are mean appropriateness scores and are calculated as described in Section 6.2. The mean appropriateness scores are shown in Table 1(b) and represented in Figure 4(b). The pairwise comparison of solutions can be seen in Figure 5(b). The comparative analysis and correction for multiple condition comparisons were performed the same way as presented in Section 6.2.

The results show that our condition (SD), with 7 other conditions, {SE, SF, SI, BM, SJ, SC, SK}, is not significantly different from a chance level performance (see dashed line in Figure 4(b)). As for the pairwise comparison, once again NA was significantly more appropriate than other conditions. Consequently, while our solution was significantly less appropriate than NA, it was significantly more appropriate than condition SH and on par with all other conditions.

It can be observed that regarding dyadic appropriateness, numerous conditions failed to be significantly different from a chance level score and, when compared to each other, they performed without significant difference. Regarding our solution, this means that despite addressing the problem in two predicting models, the generated listener behaviour was not improved compared to other approaches.

## 7 CONCLUSIONS AND TAKEAWAYS

This work presented an approach targeting a dyadic gesture generation problem utilising a Tacotron2-based solution. Based on the different behaviours an agent is expected to exhibit while speaking contrary to when it is listening, we investigated the effect of training separate models for solving this task. Our solution used the dyadic version of the model proposed by Chang et al. [9] and was trained on the speaking and listening samples of the Talking With Hands 16.2 M dataset [16]. Based on the GENEA Challenge 2023 [14] evaluation metrics, it did not perform significantly differently from the dyadic baseline and a few other conditions with regard to human-likeness, and monadic and dyadic behaviour appropriateness.

We see as a possible improvement the individual tuning of hyperparameters of the dyadic baseline model for the two distinct models we wish to produce. We believe that revising the input features of the two models would also be worthwhile. We base this on the observation that the monadic baseline (notably not using interlocutor features) performed better in the monadic appropriateness, and similarly, the dyadic baseline (using all features) performed better in the dyadic appropriateness evaluations. Perhaps if our proposed models would reflect these changes in features, or refine the current model structures based on the validation set, it could

| (a) Monadic appropriateness | | | | | | | | | (b) Dyadic appropriateness | | | | | | | | |
| Condi-tion | MAS | Pref. matched | \multicolumn Raw response count | | | | | | Condi-tion | MAS | Pref. matched | \multicolumn Raw response count | | | | | |
| | | | 2 | 1 | 0 | −1 | −2 | Sum | | | | 2 | 1 | 0 | −1 | −2 | Sum |
| NA | 0.81±0.06 | 73.6% | 755 | 452 | 185 | 217 | 157 | 1766 | NA | 0.63±0.08 | 67.9% | 367 | 272 | 98 | 189 | 88 | 1014 |
| SG | 0.39±0.07 | 61.8% | 531 | 486 | 201 | 330 | 259 | 1807 | SA | 0.09±0.06 | 53.5% | 77 | 243 | 444 | 194 | 55 | 1013 |
| SJ | 0.27±0.06 | 58.4% | 338 | 521 | 391 | 401 | 155 | 1806 | BD | 0.07±0.06 | 53.0% | 74 | 274 | 374 | 229 | 59 | 1010 |
| BM | 0.20±0.05 | 56.6% | 269 | 559 | 390 | 451 | 139 | 1808 | SB | 0.07±0.08 | 51.8% | 156 | 262 | 206 | 263 | 119 | 1006 |
| SF | 0.20±0.06 | 55.8% | 397 | 483 | 261 | 421 | 249 | 1811 | SL | 0.07±0.06 | 53.4% | 52 | 267 | 439 | 204 | 47 | 1009 |
| SK | 0.18±0.06 | 55.6% | 370 | 491 | 283 | 406 | 252 | 1802 | SE | 0.05±0.07 | 51.8% | 89 | 305 | 263 | 284 | 73 | 1014 |
| SI | 0.16±0.06 | 55.5% | 283 | 547 | 342 | 428 | 202 | 1802 | SF | 0.04±0.06 | 50.9% | 94 | 208 | 419 | 208 | 76 | 1005 |
| SE | 0.16±0.05 | 54.9% | 221 | 525 | 489 | 453 | 117 | 1805 | SI | 0.04±0.08 | 50.9% | 147 | 269 | 193 | 269 | 129 | 1007 |
| BD | 0.14±0.06 | 54.8% | 310 | 505 | 357 | 422 | 220 | 1814 | **SD** | **0.02±0.07** | **52.2%** | **85** | **307** | **278** | **241** | **106** | **1017** |
| **SD** | **0.14±0.06** | **55.0%** | **252** | **561** | **350** | **459** | **175** | **1797** | BM | −0.01±0.06 | 49.9% | 55 | 212 | 470 | 206 | 63 | 1006 |
| SB | 0.13±0.06 | 55.0% | 320 | 508 | 339 | 386 | 262 | 1815 | SJ | −0.03±0.05 | 49.1% | 31 | 157 | 617 | 168 | 39 | 1012 |
| SA | 0.11±0.06 | 53.6% | 238 | 495 | 438 | 444 | 162 | 1777 | SC | −0.03±0.05 | 49.1% | 34 | 183 | 541 | 190 | 45 | 993 |
| SH | 0.09±0.07 | 52.9% | 384 | 438 | 258 | 393 | 325 | 1798 | SK | −0.06±0.09 | 47.4% | 200 | 227 | 111 | 276 | 205 | 1019 |
| SL | 0.05±0.05 | 51.7% | 200 | 522 | 432 | 491 | 170 | 1815 | SG | −0.09±0.08 | 46.7% | 140 | 252 | 163 | 293 | 167 | 1015 |
| SC | −0.02±0.04 | 49.1% | 72 | 284 | 1057 | 314 | 76 | 1803 | SH | −0.21±0.07 | 44.0% | 55 | 237 | 308 | 270 | 144 | 1014 |

**Table 1: Summary statistics of user-study responses from both appropriateness studies (a - monadic; b - dyadic), with confidence intervals for the mean appropriateness score (MAS) at the level $\alpha = 0.05$; "Pref. matched" identifies how often test-takers preferred matched motion in terms of appropriateness after splitting ties. Conditions are ordered by MAS.**

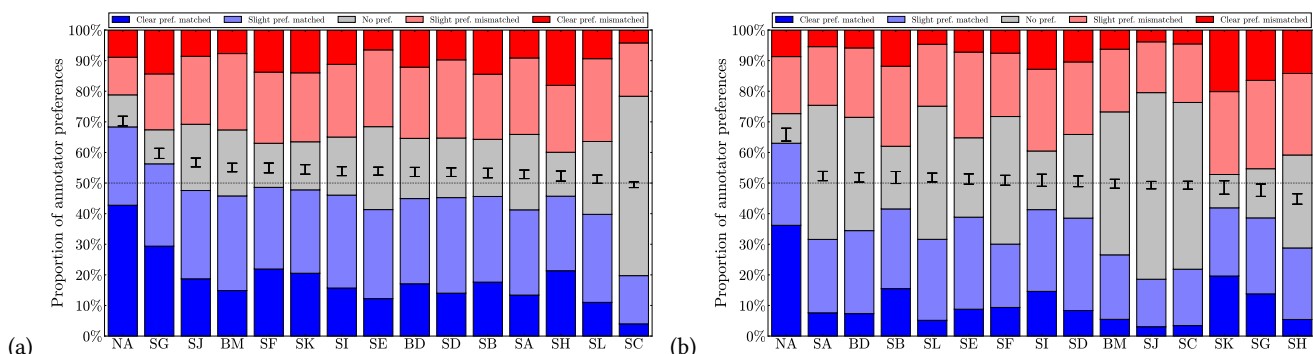

**Figure 4: Bar plots visualising the response distribution in the appropriateness studies (a - monadic; b - dyadic). The blue bar (bottom) represents responses where subjects preferred the matched motion, the light grey bar (middle) represents tied ("They are equal") responses, and the red bar (top) represents responses preferring mismatched motion, with the height of each bar being proportional to the fraction of responses in each category. Lighter colours correspond to slight preference, and darker colours to clear preference. On top of each bar is also a confidence interval for the mean appropriateness score, scaled to fit the current axes. The dotted black line indicates chance-level performance. Conditions are ordered by mean appropriateness score.**

achieve better results. Finally, we hypothesize that a conditional GAN-based (cGAN) model could improve our models' performance. Consequently, we will benchmark its performance on this dataset, and perform ablations for the splitting on the speaker and listener models. This line of thought forms the basis of our planned future work in relation to the GENEA Challenge and its dataset.

## ACKNOWLEDGMENT

This work was supported by the European Union project SERMAS and EPSRC project LISI (EP/V010875/1).

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

Received 14 July 2023; revised 11 August 2023; accepted 09 August 2023
