# OpenReview forum: "The KCL-SAIR team's entry to the GENEA Challenge 2023 Exploring Role-based Gesture Generation in Dyadic Interactions: Listener vs. Speaker"
_ACM.org/ICMI/2023/Workshop/GENEA_Challenge — GENEA Challenge 2023 Workshopproceeding_

### Official Review · Reviewer_TLGf · 2023-07-19
**Review of Role-based Gesture Generation in Dyadic Interactions**

**Rating:** 6
**Confidence:** 4

**Review:**

Paper Summary:

This paper presents a novel approach to gesture generation in dyadic interactions. The authors propose training two separate models based on the baseline model structure, one for when the agent is speaking (SM) and another for when the agent is listening (LM). The authors propose training two separate models, one for when the agent is speaking and another for when the agent is listening. The authors believe that due to the stark difference between the behavior expected from speakers and listeners, training separate models is a promising avenue. During inference, both models were loaded and outputs were generated frame by frame, selecting SM or LM depending on the speaker identity of the agent.

Relevance:

The paper is relevant to the field of gesture generation in dyadic interactions. It presents a unique approach of training two separate models for when the agent is speaking and when the agent is listening, which could potentially improve the naturalness and appropriateness of the generated gestures.

Significance:

The significance of the paper lies in its novel approach to gesture generation. By training separate models for speaking and listening, the authors aim to improve the appropriateness of the generated gestures for the respective speaker. This could potentially lead to more natural and human-like gestures in dyadic interactions.

Paper Strengths:

The paper presents a novel approach to gesture generation, which could potentially improve the naturalness and appropriateness of the generated gestures.
The authors provide a detailed description of their methodology, including the training and validation process, which could be useful for other researchers in the field.
The paper includes a thorough evaluation of the proposed models, providing insights into their performance and potential areas for improvement.

Paper Weaknesses:

The paper has several writing issues, such as missing periods in Figures 3 and 5.
The purpose of the .py file mentioned in Chapter 3 is not clear.
The effectiveness of training separate models for the speaker and listener is questionable, as theoretically, the BD should be able to learn the proposed model.
It is unclear what "In line with the training and validation practices" means, and which dataset was used for training. It appears from Figure 1 that only training data was used.

---

### Official Review · Reviewer_uw9d · 2023-07-27
**I like the idea of having speaker model and listener model**

**Rating:** 10
**Confidence:** 4

**Review:**

The paper proposed to have two models (a speaker model and a listener model) for gesture generation in dyadic situation.
The authors split dataset into a speaker dataset and a listener dataset (recognized by speaker ID) and separately trained a Tactron2-based  (for text-to-audio generation) seq2seq model. For generation, the authors generated gestures frame by frame by using the speaker model if the agent is speaking or the listener model if the agent is listening(recognized by speakerID in test data).

The paper is well organized and written.
I like the idea of having a speaker model and a listener model separately.

However, the results does not shows the effectiveness strongly. Especially, "Appropriateness for interlocutor speech"
I wonder how long time the test data have listening status and how many speeches (chunks) is the agent mainly listening in evaluation.

---

### Decision · Program_Chairs · 2023-08-07

**Decision:**

Accept (Workshop proceeding)

**Comment:**

UPDATED:

As the authors clarified ‘speaker identity’ labels, the chairs accept this paper as a workshop paper. The authors should clarify the meaning of speaker identity and how they calculated that in the camera-ready version. And please further clarify the missing details that the reviewers pointed out.


PREVIOUS:

The proposed method involves training the dyadic baseline system provided by the organisers twice, once on the “main-agent” data, and once on the “interloctr” data. The paper states that “the speaker identity labelling of
the agent was used to determine when the agent was speaking”. For generating the submitted motion, “both models were loaded [... and] outputs were generated frame by frame, selecting [one of the two trained models] depending on the speaker identity of the agent as described above”.

Unfortunately, the meta-review identified two critical flaws in the core contribution of the paper.

First, the speaker identity label does not determine whether the person is speaking or not - there is turn taking in the dataset, and the dataset was also augmented in such a way that each conversation appeared twice, with the speaker identities flipped. Therefore, the two models are trained on identical data, and should be identical.

Second, in all test input files, the speaker identity of the agent with missing motion was “main-agent”, therefore the logical conclusion is that only one of the two models was used for synthesis. This is in contrast with the main idea of the method, which is the combination of two models.

The chairs hold the decision until the authors clarify this (by email).